# Unraveling ADHD: genes, co-occurring traits, and developmental dynamics

Catriona J Miller[1], Evgeniia Golovina[1], Sreemol Gokuladhas[1], Joerg S Wicker[2] , Jessie C Jacobsen[3,4], Justin M O'Sullivan[1,5,6,7,8]

**Attention-deficit/hyperactivity disorder (ADHD) is a heterogeneous neurodevelopmental condition with a high prevalence of co-occurring conditions, contributing to increased difficulty in long-term management. Genome-wide association studies have identified variants shared between ADHD and co-occurring psychiatric disorders; however, the genetic mechanisms are not fully understood. We integrated gene expression and spatial organization data into a two-sample Mendelian randomization study for putatively causal ADHD genes in fetal and adult cortical tissues. We identified four genes putatively causal for ADHD in cortical tissues (fetal: *ST3GAL3*, *PTPRF*, *PIDD1*; adult: *ST3GAL3*, *TIE1*). Protein–protein interaction databases seeded with the causal ADHD genes identified biological pathways linking these genes with conditions (e.g., rheumatoid arthritis) and biomarkers (e.g., lymphocyte counts) known to be associated with ADHD, but without previously shown genetic relationships. The analysis was repeated on adult liver tissue, where putatively causal ADHD gene *ST3GAL3* was linked to cholesterol traits. This analysis provides insight into the tissue-dependent temporal relationships between ADHD, co-occurring traits, and biomarkers. Importantly, it delivers evidence for the genetic interplay between co-occurring conditions, both previously studied and unstudied, with ADHD.**

## Introduction

Attention-deficit/hyperactivity disorder (ADHD) is a heterogeneous neurodevelopmental condition characterized by hyperactivity, inattention, and impulsivity (Weissenberger et al, 2017; Sayal et al, 2018). There is a high prevalence of co-occurring conditions with ADHD (Pehlivanidis et al, 2020). Common co-occurring conditions include anxiety, depression, and bipolar disorder (Leitner, 2014; Pehlivanidis et al, 2020).

ADHD has a heritability of 74% with approximately a third of this heritability coming from common variants (e.g., single nucleotide polymorphisms [SNPs]) with small individual effect sizes (Faraone & Larsson, 2019). The most recent genome-wide association study (GWAS) identified 27 genetic loci associated with ADHD (Demontis et al, 2023). Consistent with the co-occurrence of conditions with ADHD, GWAS have estimated that 84–98% of ADHD-associated variants are shared with other psychiatric disorders (Demontis et al, 2019, 2023). However, as is observed for other complex polygenic traits, many of the identified SNPs are in noncoding regions of the genome and their impact(s) on both the development of ADHD and interaction(s) with co-occurring traits is(are) unclear (Zhang & Lupski, 2015; Liu et al, 2021). The ADHD-associated SNPs may mark regulatory regions, or expression quantitative trait loci (eQTLs), which spatially interact with their target genes in a temporal- and tissue-specific manner (Zhang & Lupski, 2015; GTEx Consortium, 2020). Spatial eQTLs can regulate genes in a cis (within 1 Mb of the gene), trans-acting intrachromosomal (further than 1 Mb from the gene), or trans-acting interchromosomal manner. Here, we asked the question, can we identify causal ADHD genes by integrating them into analyses of fetal and adult brain-specific chromatin interactions (Schmitt et al, 2016; Won et al, 2016) and eQTL data (Walker et al, 2019; GTEx Consortium, 2020)? Doing so will enable an understanding of individual risk of developing ADHD and its co-occurring traits.

To investigate the causality of loci associated with ADHD from GWAS, two-sample Mendelian randomization (2SMR) studies use summary-level exposure and outcome data with genetic variants as instrument variables (Hartwig et al, 2016; Dang et al 2022). Although 2SMR is increasing in popularity (Hemani et al, 2018; Woolf et al, 2022), little work has been done on combining this method with network analyses. We have previously combined our network analysis with 2SMR to understand autism (Miller et al, 2023). However, the putatively causal genes identified were noncoding, meaning that their roles were unable to be analyzed further without significant functional work.

[1]The Liggins Institute, The University of Auckland, Auckland, New Zealand   [2]School of Computer Science, University of Auckland, Auckland, New Zealand   [3]School of Biological Sciences, The University of Auckland, Auckland, New Zealand   [4]Centre for Brain Research, The University of Auckland, Auckland, New Zealand   [5]The Maurice Wilkins Centre, The University of Auckland, Auckland, New Zealand   [6]Garvan Institute of Medical Research, Sydney, Australia   [7]MRC Lifecourse Epidemiology Unit, University of Southampton, Southampton, UK   [8]Singapore Institute for Clinical Sciences, Agency for Science Technology and Research (A*STAR), Singapore, Singapore

Correspondence: justin.osullivan@auckland.ac.nz

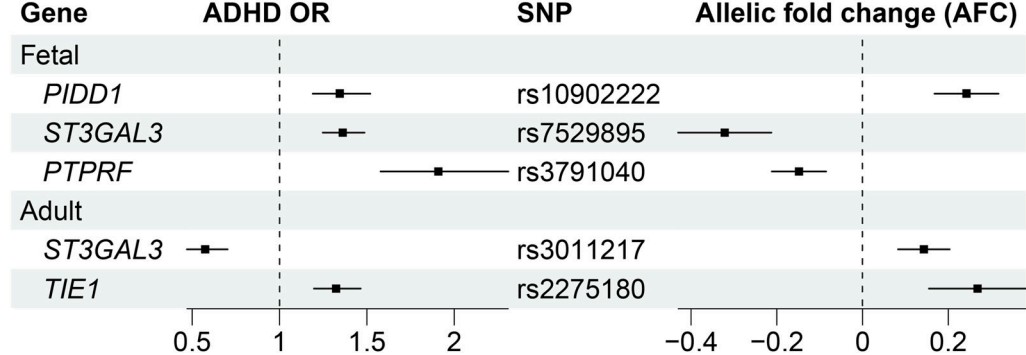

**Figure 1. Two-sample Mendelian randomization identifies three putatively causal ADHD genes in the fetal cortical tissue and two putatively causal ADHD genes in the adult cortical tissue.**
Only those statistically significant ($P <$ 0.05) after the Bonferroni correction are shown. Odds ratios (OR) are calculated based on the SNP–gene pair. The difference in odds ratios between *ST3GAL3* in the fetal and adult tissue is due to the opposing allelic fold changes (i.e., the decreased allelic fold change from rs7529895 causes an increased risk of ADHD, whereas the increased allelic fold change from rs3011217 causes a decreased risk of ADHD).

Here, we analyzed the biological pathways involving genes regulated by ADHD-associated variants and our identified putatively causal ADHD genes. We combined fetal and adult brain-specific chromatin interactions, eQTL data, and ADHD-associated variants to identify genetic associations with co-occurring traits. Expanding this work to include causal genes and protein interaction networks identified pathways connecting ADHD causal genes to biomarkers and co-occurring conditions. This analysis allowed us to further understand the interplay between ADHD and its causal genes in a tissue-specific and time-dependent manner.

# Results

### Two-sample Mendelian randomization identified causal ADHD genes within fetal and adult cortical tissues

Three genes (*PIDD1*, *ST3GAL3*, and *PTPRF*) were identified as putatively causal in the fetal cortical tissue, and two genes (*ST3GAL3* and *TIE1*) were identified as putatively causal in the adult cortical tissue using 2SMR (Figs 1 and S1, Table S1). *PIDD1* and *TIE1* are putatively causal as the increased allelic fold change, from the 2SMR instrument SNP, results in an increased odds ratio (OR) for ADHD (*PIDD1*, OR 1.34 [1.19-1.51]; *TIE1*, OR 1.32 [1.20-1.46]). A decreased allelic fold change of *ST3GAL3* and *PTPRF* results in an increased OR for ADHD (*PTPRF*, OR 1.91 [1.58–2.31]). Different instrument SNPs were identified for *ST3GAL3* in the fetal and adult cortical tissue analysis. The decreased allelic fold change of rs7529895 (*ST3GAL3*) in the fetal cortical tissue results in an increased OR for ADHD (OR 1.36 [1.25–1.49]), whereas an increased allelic fold change of rs3011217 (*ST3GAL3*) results in a decreased OR for ADHD (OR 0.57 [0.47–0.70]). These SNPs are separated by 61,395 bp on chromosome 1. Notably, three of these genes (*ST3GAL3*, *PTPRF*, and *TIE1*) are located on chromosome 1p31, within 650 kb of each other. All three genes are regulated by cis eQTLs, which were not in LD with each other, a requirement for inclusion in 2SMR. The 1p31 locus has previously been shown to be associated with psychiatric disorders (Monistrol-Mula et al, 2025) and ADHD specifically (Hall et al, 2021; Demontis et al, 2023; Deng et al, 2024).

### Fetal and adult protein–protein interaction networks (PPINs) identified genetic loci enriched for ADHD and other traits

We queried the fetal and adult cortical tissue GRNs with ADHD-associated variants (Tables S2 and S3). A mixture of neurological and non-neurological GWAS traits were identified as associated with ADHD in both the fetal and adult cortical tissue (Table S4), including many related to schizophrenia and cognition (Fig 2A). Traits appearing in the fetal cortical tissue analysis that did not appear in the adult cortical tissue included metabolic traits (e.g., HDL cholesterol and trans fatty acid levels; Fig 2B). 24 of 47 traits were brain-related (e.g., mood traits, white matter microstructure, and depression) in the adult analysis; notably, these traits were not significant in the fetal analysis (Figs S2 and S3).

Within the fetal analysis, genes located in chr17q21.31, such as *MAPT-AS1*, *KANSL1*, *LINC02210*, *MAPK8P1P2*, and *RP11-259G18.1*, were regulated by eQTLs that have known associations with cognition, schizophrenia, and alcohol consumption traits (Fig 2C, Table S5). Metabolic and fatty acid–related traits were associated with eQTLs regulating *FADS1* and *FADS2*. A cluster of genes at chr3p21.1 (*NEK4*, *GNL3*, *STAB1*, *ITIH4*, *TMEM110*) were regulated by eQTLs associated with intelligence and cognition (Figs 2C and S2).

Within the adult cortical analysis, *KANSL1* and *LINC02210* (chr17q21.31 cluster identified in the fetal analysis) were associated with eQTLs that were associated with brain-related traits (e.g., mood traits and Parkinson's disease; Table S6). However, in contrast to the fetal cortical results, we observed limited gene clustering within the adult cortical analysis. Rather, the intersection of ADHD with co-occurring traits involved apparently individual genes (Fig S3). Despite this, many of these genes were regulated by eQTLs associated with educational attainment, cognition, and schizophrenia.

### Developmental enrichment analysis identified enrichment of fetal and adult ADHD-associated genes in the dorsolateral prefrontal cortex (DLPFC)

Developmental stage– and region-specific enrichment of genes spatially regulated by ADHD-associated SNPs was identified using Allen Brain Atlas (ABA) RNA-seq data (Hawrylycz et al, 2012; Miller et al, 2014). Developmental effect scores were calculated for the

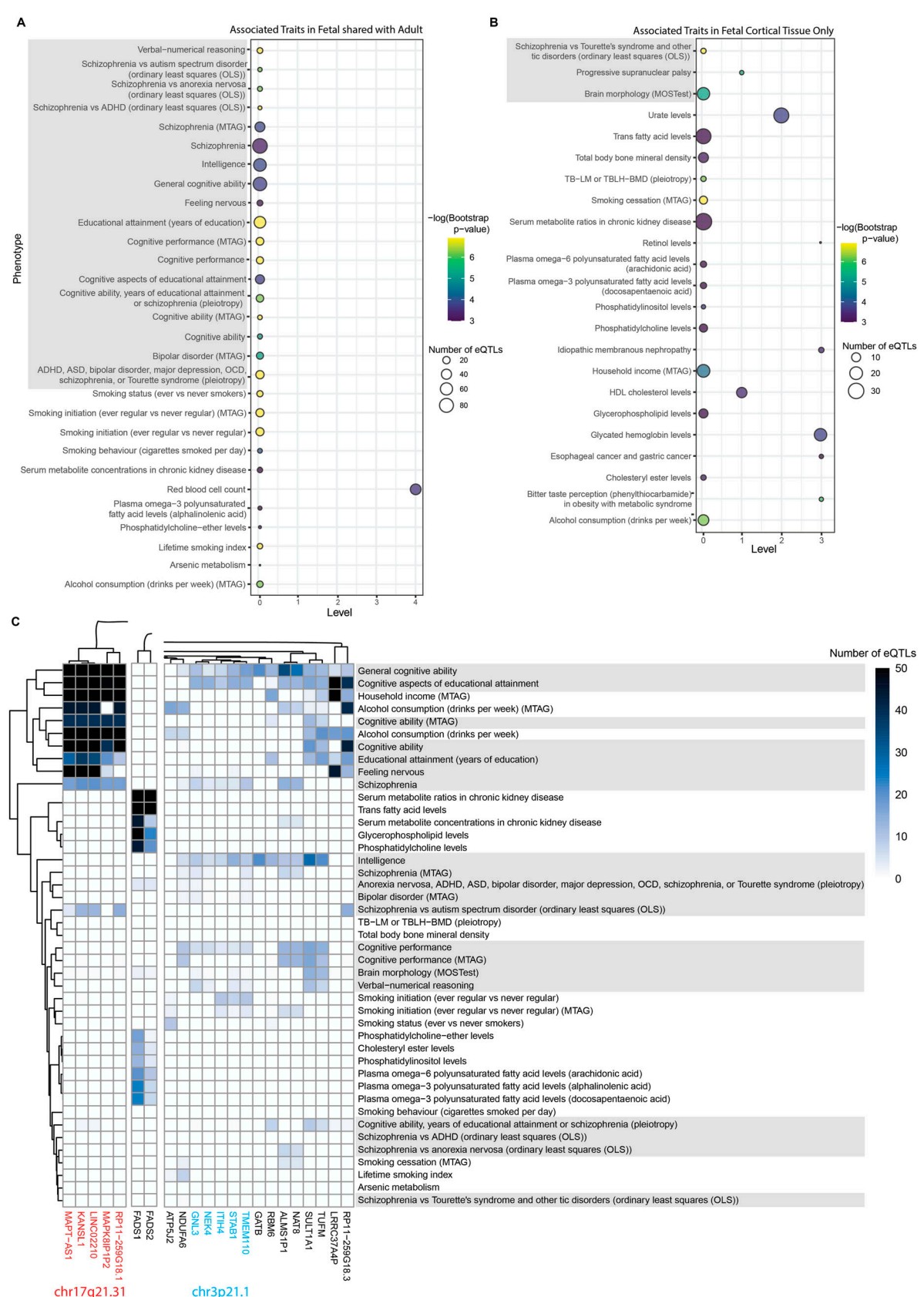

fetal and adult gene sets in 27 brain regions. In both gene sets, the DLPFC was identified as having a statistically significant developmental effect (FWER < 0.05; Table S7). The adult gene set also had a statistically significant developmental effect within the medial prefrontal cortex (Table S7). The DLPFC was enriched with the expression of the fetal gene set during infant, child, adolescent, and adult stages of development (Fig S4; Table S8). Notably, enrichment of the adult gene set within the DLPFC was only statistically significant during the adult stage of development.

### PPINs of causal genes identified novel genetic pathways linking ADHD and known co-occurring conditions

We queried the fetal and adult cortical tissue GRNs with the fetal and adult putatively causal genes, respectively, to identify the traits that co-occur with ADHD. Across both analyses, traits already known to be associated with ADHD such as lipoprotein (a) levels, lymphocyte counts, and rheumatoid arthritis (Ugur et al, 2018; Jones et al, 2019; Önder et al, 2021; Xu et al, 2021; Chiu et al, 2022) were identified (Table S9). Brain/mood-related traits (e.g., depressed affect and cortical surface area) were identified by the fetal causal analysis only (Fig 3A and B).

The protein interactions connecting the putatively causal genes (level 0) with the genes regulated by eQTLs associated with the lipoprotein (a) levels, lymphocyte counts, eye traits, and rheumatoid arthritis were identified (Fig 3C–E). The causal gene *TIE1* encodes a protein that plays a role in vascular development and function (Xu et al, 2022). Notably, through this, *TIE1* is indirectly linked to *PLG*, a gene regulated by eQTLs associated with lipoprotein (a) levels (Fig 3C, Tables S10 and S11).

Similarly, *PIDD1* is putatively causal for ADHD and encodes a protein involved in an apoptosis (cell death) network. Through this network, *PIDD1* links indirectly to *TRAF1*, which, in both whole blood and fetal cortical tissues, is regulated by SNPs associated with rheumatoid arthritis (Fig 3E, Tables S10 and S12). Notably, rs10902223 is an eQTL for both *PIDD1* and *PNPLA2*, which is associated with eye traits.

### Metabolic traits identified in the liver gene regulatory network are linked to ADHD causal genes

To further understand the metabolic pathways linked to ADHD, we ran the Multimorbid3D pipeline using a liver GRN. No causal genes were identified by 2SMR analysis using the adult liver GRN as the exposure data and the ADHD GWAS as the outcome data (Demontis et al, 2019). Therefore, we used cortical causal genes (both fetal and adult) linked to co-occurring metabolic traits from the previous analysis to run Multimorbid3D with an adult liver GRN (Tables S13 and S14). This analysis identified two brain/mood-related GWAS

traits (Alzheimer's disease and feeling fed-up) and many traits related to cholesterol metabolism (Fig 4A). Notably, the cholesterol traits on level 2 of the network are associated with SNPs that regulate *ST3GAL4* within the liver GRN (Fig 4B).

## Discussion

Two-sample Mendelian randomization analysis identified four genes (i.e., *ST3GAL3*, *PIDD1*, and *PTPRF* in the fetal cortical tissue; and *ST3GAL3* and *TIE1* in the adult cortical tissue) as being putatively causal for ADHD. A de novo network analysis of the causal genes identified potential pathways linking them to biomarkers (e.g., lipoprotein (a) levels, lymphocyte counts [Ugur et al, 2018; Önder et al, 2021; Xu et al, 2021]) and co-occurring traits (e.g., rheumatoid arthritis and eye conditions [DeCarlo et al, 2016; Jones et al, 2019; Chiu et al, 2022; Bellato et al, 2023]). Similarly, a network analysis of ADHD-associated genetic variants identified genomic regions (e.g., chr17q21.31, *FADS1/FADS2*, chr3p21.1) at the intersection of ADHD with co-occurring traits (e.g., schizophrenia, metabolic traits, and intelligence).

The methods and datasets used in this study are the source of both the strengths and limitations of the analyses. 2SMR is a popular computational technique (Hemani et al, 2018; Woolf et al, 2022) for assigning causality and does not rely on experimental analysis. However, this means that 2SMR requires the satisfaction of three key assumptions. The genetic variants (SNPs) (1) must be associated with the exposure data (e.g., fetal and adult GRNs), (2) must not be associated with any confounding factors, and (3) must not be associated with the outcome (e.g., ADHD) directly (Davies et al, 2018). The first assumption is satisfied by selecting all SNPs with an exposure $P$-value $< 1 \times 10^{-5}$. Assumptions (2) and (3) are much harder to prove. However, the use of genetic variants naturally provides a random assignment within both control and case groups, which should reduce the impacts of confounding. Sensitivity analyses, including a Cochrane Q test and an MR–Egger pleiotropy test, were also performed to remove horizontal pleiotropy and therefore reduce potential biases associated with assumptions (2) and (3). Therefore, we refer to our identified genes as putatively causal for ADHD until empirically proven. Similarly, Multimorbid3D relies upon the supposedly simple assumption that the identified proteins form characterized protein–protein interactions. However, this means that genes encoding proteins that form unknown, or do not form (e.g., long noncoding RNAs), protein–protein interactions cannot be included in the analysis. Only common SNPs were included in this analysis as we focused on variants identified as associated with ADHD through GWAS. It would be interesting to expand this work to include CNVs; however, this would require different methodologies than using spatial eQTLs

**Figure 2. Multimorbid3D identifies traits associated with ADHD in the fetal cortical tissue.**
**(A)** Bubble plot showing statistically significant traits (bootstrap $P < 0.05$) found in the analysis of ADHD GWAS variants in the fetal cortical tissue that were also identified in the adult cortical tissue. The size of the bubbles refers to the number of eQTLs associated with that particular trait. Bubble colors refer to the $-\log(P$-value) after bootstrapping. The y-axis trait names come from the GWAS Catalog, and brain/mood-related traits are highlighted gray. **(B)** is the same as (A) for traits identified in the fetal analysis only. **(C)** Heatmap outlining which genes are regulated by variants associated with the traits from (A, B) in the fetal cortical tissue. The color is based on the number of eQTLs associated with the trait and gene. Clustering is based on the numbers of eQTLs. The first subsection is genes at chr17q21.31, the second subsection is *FADS1/FADS2*, and the third subsection is genes related to cognition, including those at chr3p21.1. The full heatmap is shown in Fig S1.

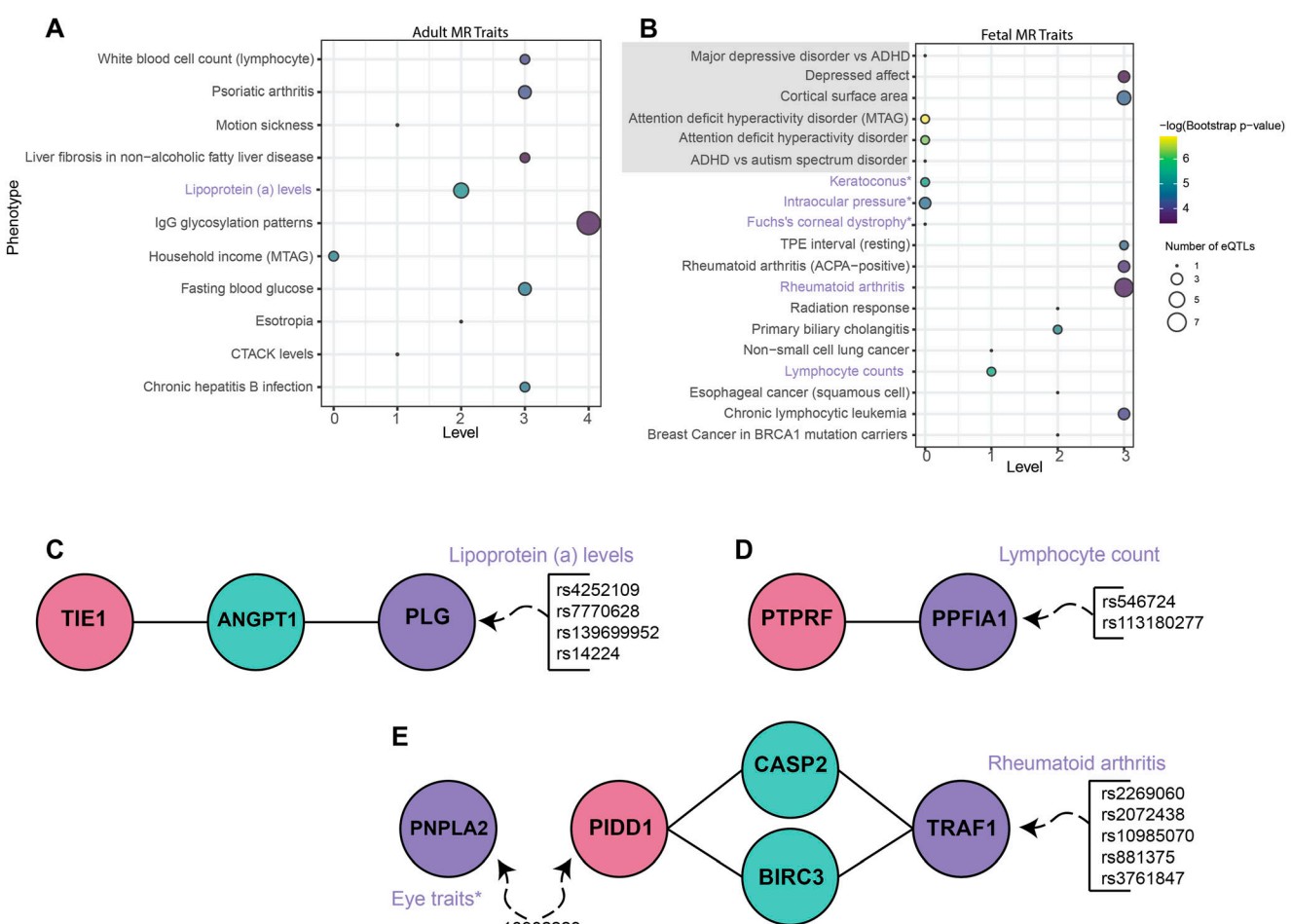

**Figure 3. Multimorbid3D on causal genes identifies traits co-occurring with ADHD.**
**(A)** Bubble plot showing statistically significant traits (bootstrap P < 0.05) found in the analysis of causal ADHD genes in the adult cortical tissue. The size of the bubbles refers to the number of eQTLs associated with that particular trait. Bubble colors refer to the −log(P-value) after bootstrapping. Y-axis trait names come from the GWAS Catalog, and brain/mood-related traits are highlighted gray. **(B)** is the same as (A) for causal ADHD genes in the fetal cortical tissue. Purple traits in (A, B) are related to the networks in (C, D, E). **(C, D)** highlights the protein network linking the adult causal gene *TIE1* to lipoprotein (a) levels, whereas (D) links the fetal causal gene *PTPRF* to lymphocyte counts; both potential biomarkers. **(E)** shows the protein pathways linking the fetal causal gene *PIDD1* to co-occurring conditions related to the eye and rheumatoid arthritis. In (C, D, E), the red bubble is for proteins encoded for by causal genes, green is for proteins in the network, and purple is for proteins encoded for by genes regulated by SNPs associated with the listed trait.

within the GRNs to elucidate putatively causal mechanisms. Rare SNPs were also ignored as a larger dataset would be required to identify putatively causal genes through 2SMR. It remains possible that rare variants contribute to the regulation of the identified genes; however, further work would need to be done on this. Despite these limitations, a strength of this study lies in the integration of datasets across many biological levels (i.e., fetal and adult cortical tissue-specific 3D genome structure, GWAS SNPs, gene expression data, and protein–protein interactions) to identify the genetic links and potential mechanisms between ADHD and co-occurring traits.

ADHD is widely accepted to be a heterogeneous condition. However, the co-occurring traits are "all" recognized as being at least partially dependent upon inherited germ line variation. Therefore, it was reassuring to identify the chr17q21.31, chr3p21.1, and *FADS1/FADS2* loci, which are associated with schizophrenia, cognition, and household income; cognition; and metabolic

traits, respectively, as linking these co-occurring traits to ADHD. We previously identified the chr17q21.31 locus as contributing to the intersection of autism with neurological traits (e.g., Parkinson's disease and white matter microstructure) (Miller et al, 2023). The chr17q21.31 locus has been identified as an autism susceptibility locus (Pain et al, 2019), whereas the chr3p21.1 locus has been linked to psychiatric conditions and cognition (Yang et al, 2020). We contend that the similarities between the results for ADHD and autism reflect the commonalities between these two neurodevelopmental conditions. This is supported by the large overlap between the two conditions and the fact that autism was previously an exclusion criterion for an ADHD diagnosis (Leitner, 2014).

The identification of the DLPFC as being enriched in the ADHD-associated fetal and adult cortical gene sets is both significant and expected given the region's known role in ADHD (Nejati et al, 2021; Wu et al, 2023). However, the time-dependent enrichment is an

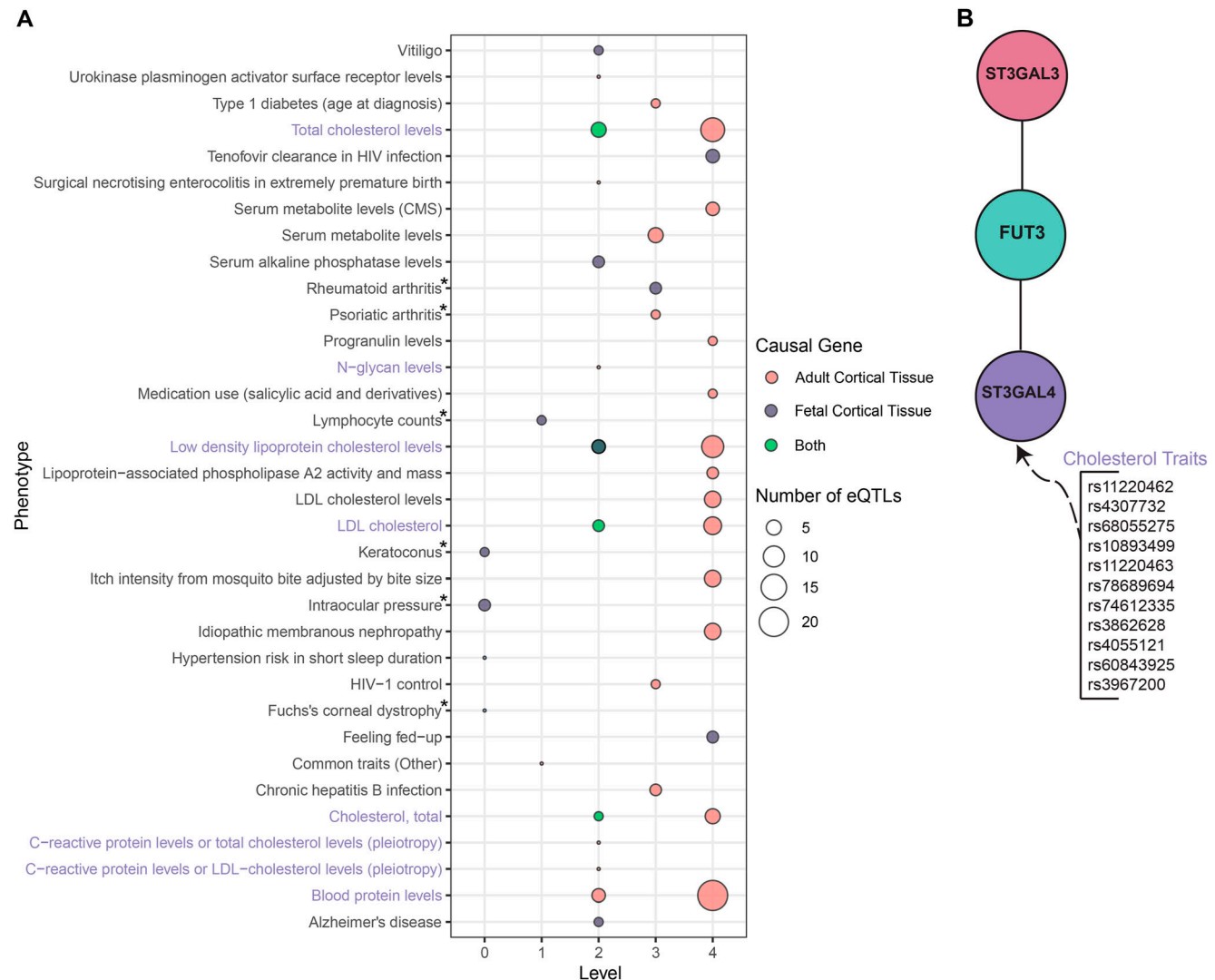

**Figure 4. ADHD causal genes associated with cholesterol traits in the liver GRN.**
**(A)** Bubble plot of statistically significant traits (bootstrap *P* < 0.05) identified in the analysis of causal ADHD genes in the liver tissue. The size of the bubbles refers to the number of eQTLs associated with that particular trait. Bubble colors refer to whether the trait comes from the analysis of genes from the fetal causal genes (*ST3GAL3*, *PTPRF*, *PIDD1*) or the adult causal genes (*ST3GAL3*, *TIE1*), or whether it was present in both analyses. Asterisk traits were present in the cortical tissue analysis. **(B)** shows the network genetically linking the causal gene *ST3GAL3* to cholesterol traits highlighted purple in (A). Coloring of (B) matches Fig 3C–E.

important step toward identifying the key developmental windows that link ADHD with certain co-occurring traits or biomarkers. This knowledge could enable targeted therapeutic management of the genetic pathways linking ADHD and the co-occurring traits. For example, the apparent significance of the fetal gene set from early timepoints is not only consistent with ADHD being a childhood-onset condition, but also suggests biological processes that may be therapeutically targeted in populations at risk of associated co-occurring traits to reduce an individual's chance of developing these co-occurring traits. The enrichment of the adult cortical gene set in only the adult timepoint provides a potential argument for adult-onset ADHD; a highly debated idea with further research needed (Moffitt et al, 2015; Faraone & Biederman, 2016; Taylor et al, 2022).

We previously identified four noncoding genes as being putatively causal for autism (Miller et al, 2023). In contrast, our current study identified four protein-coding genes (i.e., *ST3GAL3*, *PIDD1*, and *PTPRF* in the fetal cortical tissue; and *ST3GAL3* and *TIE1* in the adult cortical tissue) that are putatively causal for ADHD. We identified reductions in the expression of *ST3GAL3* as being associated with an increased risk of developing ADHD. Similarly, we demonstrated that an increase in *ST3GAL3* expression because of a different eQTL was protective against ADHD. Our results support the findings of Rivero et al (2021) who identified complete inactivation of *ST3GAL3* as leading to profound intellectual disability and predicted that more subtle changes in gene expression could increase an individual's chance of developing ADHD. *ST3GAL3* was also one of the strongest loci identified in the Demontis et al (2019) GWAS.

Analyzing the protein–protein interactions of the proteins encoded by the causal genes informs hypothesis development for the potential mechanisms that link ADHD with co-occurring traits and biomarkers. However, future work must empirically test the validity of the predictions from these hypotheses. For example, investigating *PIDD1* in the fetal cortical tissue identified links with eye conditions and rheumatoid arthritis, both previously associated with ADHD (DeCarlo et al, 2016; Jones et al, 2019; Chiu et al, 2022; Bellato et al, 2023). *PIDD1* has been linked to ADHD through promoter methylation, and an increased expression of *PIDD1* has been observed in brain tissues from individuals with ADHD (Pineda-Cirera et al, 2019). This is consistent with our 2SMR results, which identified that an increase in *PIDD1* expression resulted in an increased OR for ADHD. *PIDD1* is regulated in the fetal cortical tissue by rs10902223, which has been linked with eye traits (i.e., keratoconus and intraocular pressure [McComish et al, 2020]). rs10902223 falls within and regulates *PNPLA2* in whole blood, which encodes an essential enzyme for the maintenance of vision (Hara et al, 2023). We contend that within the fetal cortical tissue, rs10902223 may impact on ADHD via *PIDD1*, while also impacting eye conditions through its action on *PNPLA2* expression. *PIDD1* was associated with rheumatoid arthritis through the network connecting it with *TRAF1*. This network was shown (STRING; https://string-db.org [Szklarczyk et al, 2019]) to be involved in apoptosis, which has previously been linked to brain size in individuals with ADHD through MRI and microarray data (Hess et al, 2021). *TRAF1* is also regulated by SNPs associated with rheumatoid arthritis in both the fetal cortical tissue and the liver, and a decreased expression of *TRAF1* has been associated with increased inflammation (Abdul-Sater et al, 2017; Edilova et al, 2018). Therefore, the apoptosis network involving *PIDD1* may impact ADHD in the brain and rheumatoid arthritis in the liver.

Analyzing causal networks can inform on potential biomarkers for ADHD. For example, *TIE1* and *PTPRF* were associated with lipoprotein (a) levels and lymphocyte counts, respectively, through their protein networks. *TIE1*, while linked with abstract thinking in ADHD, plays a role in vascular development and function, with overexpression potentially increasing the risk of neuro-inflammation and destruction of the blood–brain barrier (Xu et al, 2022). Through protein interactions, *TIE1* is associated with *PLG*, which is regulated by SNPs that are associated with levels of lipoprotein (a), a form of low-density lipoprotein. Significantly higher levels of low-density lipoprotein have been identified in individuals with ADHD (Ugur et al, 2018; Xu et al, 2021).

Önder et al (2021) suggested that inflammatory responses within ADHD were related to the close relationship between ADHD and immune traits, something that we have seen in our analysis, particularly with the link to the autoimmune condition rheumatoid arthritis. However, it is unknown which inflammatory-related genes are responsible for this relationship between inflammation and ADHD (Leffa et al, 2019; Önder et al, 2021). The causal gene *PTPRF* has been linked to inflammation through increasing cytokine levels (Huang et al, 2022). PTPRF protein interacts with PPFIA1 (liprin-alpha-1), which is regulated by SNPs associated with lymphocyte count in both the fetal cortical tissue and whole blood. Neutrophil–lymphocyte ratio is significantly higher in individuals with ADHD and has been identified as a potential biomarker for inflammatory mechanisms related to ADHD (Önder et al, 2021).

Many protein tyrosine phosphatases, including *PTPRF*, are differentially expressed in immune cells compared with nonimmune cells (Arimura & Yagi, 2010). Loss of PTPRF in mice has been shown to cause hyperactivation of plasmacytoid dendritic immune cells and mild colitis (Bunin et al, 2015). ADHD has been linked to the immune system previously, through both immune-related genes and the co-occurrence of immune conditions such as allergies and autoimmune diseases (Cortese et al, 2018; Hoekstra, 2019). Therefore, we contend that the ADHD causal gene *PTPRF* and the interactions between proteins PTPRF and PPFIA1 may be important factors in the interplay between inflammation, the immune system, and ADHD.

Metabolic traits were identified as co-occurring with ADHD in both our network analyses and previous studies (Chen et al, 2018; Huber et al, 2023). However, we did not identify any causal ADHD genes in the liver GRN. This finding supports the hypothesis that ADHD is a neurodevelopmental condition with metabolic co-occurring traits, not the opposite. Despite this, the causal genes are associated with many cholesterol-related traits through pathways in the liver. This is consistent with epidemiological results, which have identified a strong link between cholesterol levels and ADHD (Ugur et al, 2018; Xu et al, 2021). Similarly, individuals with ADHD have a heightened risk (OR = 1.96) of developing cardiovascular diseases (Li et al, 2023). In our analysis, we did not identify any specific co-occurring cardiovascular diseases but instead focused on cholesterol. For example, García-Marín et al (2021) identified genetic variants associated with low HDL levels and an increased ADHD risk. However, the biological mechanisms that intertwine cholesterol metabolism and ADHD remain unknown. Here, we have identified a potential mechanism for their association through the causal gene *ST3GAL3*. In the fetal and adult cortical tissue, we have shown *ST3GAL3* to be linked with educational attainment and cognition in individuals with ADHD, whereas in the liver, it is associated with *ST3GAL4*, which is regulated by many genetic variants associated with cholesterol traits. Therefore, it is proposed that within the brain, *ST3GAL3* is related to the cognitive traits linked with ADHD, whereas in the liver, it is associated with the cholesterol co-occurring traits.

The findings from our causal and associated analyses provide a greater understanding of the biological mechanisms linking ADHD with co-occurring traits and the time dependency of these interactions. Through our analysis, we found the DLPFC to be enriched in genes, regulated by ADHD-associated SNPs, that form clusters at the intersection of ADHD with co-occurring traits. We have identified putatively causal ADHD genes and shown how interactions with proteins encoded by these genes could link ADHD with both biomarkers and co-occurring conditions. Identified biomarkers and impacted protein pathways could be used for diagnosis and stratification of individuals with ADHD based on their co-occurring traits. Future work should look at providing functional evidence to confirm these hypotheses. Evidence could come from targeted CRISPR of our identified loci in pluripotent stem cells (Wang et al, 2017). After differentiating into neurons, transcriptional analysis would be done to show functional connections between the eQTLs and putatively causal genes. Targeting the genes as knockdowns in animal models and testing for traits associated with ADHD, as well as the identified co-occurring traits, would provide evidence for the

putative causality. This work provides a step toward a time-dependent and tissue-specific understanding of how ADHD-related biological pathways contribute to an individual's overall phenotype, including the interplay with other traits.

# Materials and Methods

### Fetal and adult cortical tissue gene regulatory networks

Fetal and adult cortical tissue gene regulatory networks created in a previous study were used for this analysis (fetal, Miller [2023a]; adult, Miller [2023b]; [Miller et al, 2023]). These used eQTL databases (Walker et al, 2019; GTEx Consortium, 2020) and Hi-C chromatin libraries (Schmitt et al, 2016; Won et al, 2016) to generate a list of spatial SNP–gene regulatory pairs (Fig S1).

### Two-sample Mendelian randomization

A 2SMR study was completed using the TwoSampleMR R package (https://github.com/MRCIEU/TwoSampleMR/, version 0.5.6) (Hemani et al, 2018) to identify genes putatively causal for ADHD within the fetal and adult cortical tissue GRNs. The 2017 iPSYCH ADHD GWAS was used as the outcome data (Demontis et al, 2019), whereas all gene–eQTL pairings in the fetal or adult cortical tissue GRNs were used as the exposure data (Fig S1) (Miller et al, 2023). At the time of analysis, the 2023 ADHD GWAS (Demontis et al, 2023) had just been published and was not easily able to be integrated with the software. The exposure data were filtered to remove all eQTLs with an exposure $P$-value > $1 \times 10^{-5}$. This satisfies the 2SMR assumption that the genetic instruments (i.e., eQTLs) must be associated with the exposure (i.e., gene expression). The exposure data were then clumped to ensure all exposure instruments were independent (Hemani et al, 2018). TwoSampleMR performs LD clumping using PLINK (Purcell et al, 2007; Hemani et al, 2018) with a clumping window of 10,000 kb and an $r^2$ cut-off of 0.001. The outcome GWAS was downloaded from the IEU Open GWAS Project (Elsworth et al, 2020 *Preprint*) within the TwoSampleMR package. Harmonization of the exposure and outcome data ensured that the direction of any allele effects was the same between both datasets. Genes with two or more eQTLs (i.e., instruments) underwent sensitivity analyses to test for horizontal pleiotropy. A Cochrane Q test was performed using TwoSampleMR's *mr_heterogeneity()* function, and all genes with significant heterogeneity ($P < 0.05$) were removed. Genes with significant horizontal pleiotropy were identified using TwoSampleMR's *mr_pleiotropy_test()* function. Genes (i.e., exposures) with a $P$-value < 0.05 have a significant nonzero MR–Egger regression y-intercept, indicative of horizontal pleiotropy. Therefore, genes with a $P$-value < 0.05 were removed. Genes with one eQTL pairing underwent 2SMR using the Wald test, whereas genes with multiple eQTL pairings underwent 2SMR using MR–Egger regression and inverse variance weighted methods (Dang et al, 2022). To correct for multiple tests, a Bonferroni correction was used (adjusted $P < 0.05$). Any genes with an adjusted $P$-value below this threshold were considered putative causal ADHD genes in the fetal or adult cortical tissue. A STROBE-MR checklist was completed (Table S15) (Skrivankova et al, 2021a; Skrivankova et al, 2021b).

### Definition of ADHD-associated SNPs

SNPs associated with ADHD were downloaded from GWAS (www.ebi.ac.uk/gwas; 2023/04/06). A keyword search was done for "ADHD" on the GWAS Catalog, and all those (1,391 SNPs) with a $P$-value $5 \times 10^{-8}$ were selected.

### Identification of ADHD-associated traits

The Multimorbid3D pipeline (Golovina et al, 2023) (https://github.com/Genome3d/multimorbid3D) was used to identify traits associated with ADHD in the fetal and adult cortical tissue separately (Fig S1). The 1,391 SNPs associated with ADHD, as well as those within linkage disequilibrium ($r^2 = 0.8$; width = 5,000 bp; 7,898 SNPs in total), were input into this pipeline. Multimorbid3D queried the fetal or adult cortical tissue-specific GRN (Miller et al, 2023) to identify genes that the ADHD-associated SNPs spatially regulate within that tissue (i.e., eQTL–gene pairings). These eQTL–gene pairings made up the ADHD-specific fetal or adult cortical tissue GRN. Proteins encoded for by these genes formed "level 0" of the ADHD-specific fetal or adult cortical tissue PPIN. This network was expanded to include four outer levels of proximal protein interactions using STRING (https://string-db.org). Proteins on "level 1" were those that were identified as interacting with an ADHD-associated protein on level 0. Those on levels 2–4 interacted with a protein on the previous level. Interactions were limited to species "*Homo sapiens*" and included those with an interaction score above 0.7. The protein–protein interactions were based on evidence from experiments, text mining, co-expression, and databases (Szklarczyk et al, 2019).

Genes encoding proteins on each level were used to query the fetal or adult cortical tissue GRN to identify eQTLs with a regulatory impact on them (adjusted $P < 0.05$). GWAS was then queried with these eQTLs to identify associated traits (hypergeometric test; $P < 0.05$).

Bootstrapping was completed on the whole pipeline. 1,000 iterations were completed, and at each iteration, 7,898 trait-associated SNPs were randomly selected from the GWAS Catalog to input into Multimorbid3D. A $P$-value was calculated for each associated trait by determining the proportion of bootstrap iterations where the number of trait-associated SNPs was larger than the number in our analysis (observed). If a bootstrapped $P$-value < 0.01, we assume that the observed relationship is not random.

$$pval_{bootstrapped} = \frac{\Sigma(bootrapped \geq observed)}{1000}.$$

### Developmental enrichment analysis of ADHD-associated genes

To understand which brain regions and developmental timepoints the genes regulated by the ADHD-associated SNPs were most enriched in, a developmental enrichment analysis was undertaken

using ABAEnrichment (Grote et al, 2016). This package uses RNA-seq data from 42 individuals at five developmental stages (pre-natal, infant, child, adolescent, and adult) from ABA. Using our level 0 gene list from both the fetal and adult analysis, ABAEnrichment was used to see whether specific regions or timepoints showed significant enrichment. Genes are annotated to brain regions if their expression in that region is above the cut-off, with 0.5, 0.7, and 0.9 used as expression quantile cut-offs in this analysis (Grote et al, 2016). A hypergeometric test then compares the enrichment of annotated genes from the gene list with annotated background genes (i.e., genes not from the gene list). A family-wise error rate is calculated for each enrichment by comparing against 100 random sets. To determine which of the 42 brain regions were developmentally enriched overall, the developmental effect score dataset was used, which contains gene age effect scores based on expression change during development.

### Identification of potential ADHD co-occurring traits

To identify traits associated with the putatively causal ADHD genes, the Multimorbid3D pipeline was repeated in the fetal and adult cortical tissue separately (Fig S1). The genes identified as putatively causal in the fetal and adult cortical tissue during the 2SMR analysis were used to query the fetal or adult cortical tissue–specific GRN, respectively, to identify SNPs that spatially regulate them. These eQTL–gene pairings made up the ADHD-specific causal fetal or adult cortical tissue GRN. The rest of the process was the same as that outlined for the identification of ADHD-associated traits. Bootstrapping was done on the whole pipeline as previously described.

### Creation of the liver gene regulatory network

A liver GRN was created using the CoDeS3D algorithm (Fadason et al, 2018) (https://github.com/Genome3d/codes3d). Hi-C chromatin data (Schmitt et al, 2016), derived from liver cells, were downloaded from GEO (https://www.ncbi.nlm.nih.gov/geo/, accession: GSE87112). As an input into CoDeS3D, all SNPs from the liver tissue–specific GTEx (GTEx Consortium, 2020) eQTL dataset were fed in. The Hi-C chromatin interaction data were used to identify restriction fragments containing SNPs interacting with restriction fragments overlapping genes. The liver tissue (GTEx Consortium, 2020) eQTL dataset was used to identify interacting SNPs that were eQTLs and therefore formed eQTL–gene pairings. Significant eQTL–gene pairs (adjusted $P \leq 0.05$) after the Benjamini–Hochberg multiple testing correction formed the liver GRN (Gokuladhas et al, 2021 Preprint).

### Identification of metabolic traits associated with ADHD

The previous methods were repeated using the liver GRN instead of the fetal or adult cortical tissue GRNs. A 2SMR was undertaken following the same process but using the eQTL–gene pairings in the liver GRN as exposure data.

ADHD-associated traits were identified in the liver through Multimorbid3D by querying the liver GRN with (1) ADHD-associated

SNPs including those in LD ($r^2$ = 0.8; width = 5,000 bp), (2) fetal cortical tissue putatively causal ADHD genes, and (3) adult cortical tissue putatively causal ADHD genes.

## Data Availability

The multimorbid3D pipeline was created and run in Python (version 3.8.8). All visualizations and data analysis were performed in R (version 4.2.0) through RStudio (version 2022.02.2). Table S16 lists the datasets and software that have been used in our analyses. All scripts are available on GitHub (https://github.com/Catriona-Miller/ADHD_Co-occurring_Traits).

### Ethics statement

Ethics approval was obtained from the University of Auckland Human Participants Ethics Committee (Decoding SNPs in context, UAHPEC19373).

## Supplementary Information

## Acknowledgements

We would like to thank the Genomics and Systems Biology Group (Liggins Institute, University of Auckland) for their insightful suggestions and discussions. Data from this work come from the Genotype–Tissue Expression (GTEx) Project, which was supported by the Common Fund of the Office of the Director of the National Institutes of Health, and by NCI, NHGRI, NHLBI, NIDA, NIMH, and NINDS. CJ Miller was funded by the University of Auckland Doctoral Scholarship. E Golovina, S Gokuladhas, and JM O'Sullivan are funded by the Dines Family Foundation.

### Author Contributions

CJ Miller: formal analysis, visualization, methodology, and writing—original draft, review, and editing.
E Golovina: data curation, supervision, and writing—review and editing.
S Gokuladhas: data curation and writing—review and editing.
JS Wicker: supervision, methodology, and writing—review and editing.
JC Jacobsen: supervision, methodology, and writing—review and editing.
JM O'Sullivan: conceptualization, supervision, methodology, and writing—review and editing.

### Conflict of Interest Statement

The authors declare that they have no conflict of interest.

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
