## [Reviewer comments · Life Science Alliance]

Life Science Alliance

Unravelling ADHD: Genes, Co-occurring Traits and Developmental Dynamics

Catriona Miller, Evgeniia Golovina, Sreemol Gokuladhas, Joerg Wicker, Jessie Jacobsen, and Justin O'Sullivan
DOI: <https://doi.org/10.26508/lsa.202403029>

Corresponding author(s): Justin O'Sullivan, The University of Auckland

Review Timeline:

Submission Date:	2024-09-02
Editorial Decision:	2025-01-02
Revision Received:	2025-02-09
Editorial Decision:	2025-02-11
Revision Received:	2025-02-12
Accepted:	2025-02-14

Transaction Report:

January 2, 2025

Re: Life Science Alliance manuscript #LSA-2024-03029

Dr. Justin M. O'Sullivan
The University of Auckland
The Liggins Institute
University of Auckland
Private Bag 92019
Auckland 1142
New Zealand

Dear Dr. O'Sullivan,

Thank you for submitting your manuscript entitled "Unravelling ADHD: Genes, Co-occurring Traits and Developmental Dynamics" to Life Science Alliance. The manuscript was assessed by expert reviewers, whose comments are appended to this letter. We invite you to submit a revised manuscript addressing the Reviewer comments.

Thank you for this interesting contribution to Life Science Alliance. We are looking forward to receiving your revised manuscript.

Sincerely,

Eric Sawey, PhD
Executive Editor
Life Science Alliance
<http://www.lsa-journal.org>

B. MANUSCRIPT ORGANIZATION AND FORMATTING:

Reviewer #1 (Comments to the Authors (Required)):

The paper by Miller and coworkers reports on a pertinent bioinformatics approach to identify potential ADHD risk genes and biochemical pathways involving these genes. The authors are using a combination of fetal and adult brain-specific chromatin interactions, eQTL data, and GWAS-derived ADHD-associated variants to identify genetic associations with ADHD and with co-occurring traits. In their analyses they further include protein interaction networks of potentially causal genes. The results indicate an interplay between ADHD, its comorbidity and its risk genes in a tissue-specific and time-dependent fashion.

The analytical methods appear appropriate and the paper is well written. The following suggestions may further expand and enhance the conclusions of the paper.

- The authors should comment on their rationale to include only common SNP variants but not rare variants and CNV findings.
- Three of the proposed genes (ST3GAL3, PTPRF, TIE1) cluster in a single region on chromosome 1 and are surprising close to each other (within 650 kb). This should be explicitly mentioned and the potential implications should be discussed.
- The link of the candidate genes to immunological mechanisms should be discussed in more detail with respect to the comorbid involvement of the immune system in ADHD, such as allergies, atopic dermatitis, asthma etc. Likewise the link to metabolic and cardiovascular disease should be given more attention.
- Minor points:
 - page 6, line 231: The reference should be replaced or expanded with Demontis et al. 2019.
 - page 6, line 239: promoter
 - page 9, line 383: bootstrapped

Reviewer #2 (Comments to the Authors (Required)):

Attention-deficit/hyperactivity disorder (ADHD) is a heterogeneous neurodevelopmental condition with a high prevalence of co-occurring conditions, contributing to increased difficulty in long-term management. Genome-wide association studies have identified variants shared between ADHD and co-occurring psychiatric disorders; however, the genetic mechanisms are not fully understood. This study integrated gene expression and spatial organization data into a two-sample Mendelian Randomisation study for putatively causal ADHD genes in fetal and adult cortical tissues. They identified four genes putatively causal for ADHD in cortical tissue (fetal: ST3GAL3, PTPRF, PIDD1; adult: ST3GAL3, TIE1). Protein-protein interaction databases seeded with the causal ADHD genes identified biological pathways linking these genes with conditions (e.g. rheumatoid arthritis) and biomarkers (e.g. lymphocyte counts) known to be associated with ADHD, but without previously shown genetic relationships. The analysis was repeated on adult liver tissue, where putatively causal ADHD gene ST3GAL3 linked to cholesterol traits. The study concludes that this analysis provides insight into the tissue-dependent, temporal relationships between ADHD, co-occurring traits, and biomarkers. Importantly, it delivers evidence for the genetic interplay between co-occurring conditions, both previously studied and unstudied, with ADHD. The study's key contribution to the field is that identify genes putatively causal for ADHD and show how their biological pathways link to co-occurring traits and biomarkers in a tissue and time-dependent manner.

Comments

It is a truly impressive comprehensive study using an advanced methodology, timely, and well-described, with multiple strengths, and some limitations also well detailed by the authors. Worthwhile highlighting and detailing them below. The authors are commended for undertaking this important work in studying the underlying genetics of ADHD that has a heritability of 74% with approximately a third of this heritability coming from common variants. As it is observed for other complex polygenic traits, many of the identified SNPs are in non-coding regions of the genome and their impact(s) on both the development of ADHD and interaction(s) with co-occurring traits is unclear. Thus, in this study, the authors asked the question of whether they can identify causal ADHD genes by integrating them into analyses of fetal and adult brain-specific chromatin interactions and eQTL data in order to enable an understanding of individual risk of developing ADHD and its co-occurring traits.

The authors relied on some aspects of their previously published methodology (C. Miller et al. 2023), although the putatively causal genes identified were non-coding, meaning that their roles were unable to be analyzed further. Therefore, in this study, the authors expanded the effort to include causal genes and protein interaction networks and identified pathways connecting ADHD causal genes to biomarkers and co-occurring conditions.

In contrast here, two-sample Mendelian Randomisation analysis identified four coding genes (i.e. ST3GAL3, PIDD1, and PTPRF in fetal cortical tissue; and ST3GAL3 and TIE1 in adult cortical tissue) as being putatively causal for ADHD. A de novo network analysis of the causal genes identified potential pathways linking them to biomarkers (e.g. lipoprotein (a) levels, lymphocyte counts and co-occurring traits (e.g. rheumatoid arthritis and eye conditions). Similarly, a network analysis of ADHD-associated genetic variants identified genomic regions (e.g. chr17q21.31, FADS1/FADS2, chr3p21.1) at the intersection between ADHD and co-occurring traits (e.g. schizophrenia, metabolic traits and intelligence).

To expand, there are truly multiple strengths of this approach, such as the integration of datasets across many biological levels (i.e. fetal and adult cortical tissue specific 3D genome structure, GWAS SNPs with the iPSYCH ADHD GWAS modified data used as the outcome data (Demontis et al. 2019) while all gene eQTL pairings in the fetal or adult cortical tissue GRNs were used as the exposure data (C. J. Miller et al. 2023), gene expression data, and protein-protein interactions) to identify the genetic links and potential mechanisms between ADHD and co-occurring traits.

Identification of the DLPFC as being enriched in the ADHD-associated fetal and adult cortical gene-sets is both significant and expected given the region's known role in ADHD, which they connect with the time-dependent enrichment as an important step towards identifying the key developmental windows that link ADHD with certain co-occurring traits or biomarkers. For example they contend that ADHD causal gene PTPRF and the interactions between proteins PTPRF and PPFIA1 may be important factors in the interplay between inflammation and ADHD. The knowledge could enable targeted therapeutic management of the genetic pathways linking ADHD and the co-occurring traits.

Limitations, implications, and future work

The authors also rightfully point out and discuss that the methods and datasets used in this study are the source of both the strengths and limitations of the analyses. Therefore, the authors refer to the four identified genes as putatively causal for ADHD until empirically proven.

While identified biomarkers and impacted protein pathways could be used for diagnosis and stratification of individuals with ADHD based on their co-occurring traits, a future work should look at providing functional evidence to confirm these hypotheses. Evidence could come from targeted CRISPR of this group's identified loci in pluripotent stem cells (Wang et al. 2017). After differentiating into neurons, transcriptional analysis would be done to show functional connections between the eQTLs and putatively causal genes.

Targeting the genes as knockdowns in animal models and testing for traits associated with ADHD as well as the identified co-occurring traits would provide evidence for the putative causality. This work, indeed, provides a step towards a time-dependent and tissue-specific understanding of how ADHD-related biological pathways contribute to an individual's overall phenotype, including the interplay with other traits.

Dear Sir/Madam

Thank you for the opportunity to respond to the reviewer comments on our manuscript. Please find below our responses to the comments on 'Unravelling ADHD: genes, co-occurring traits and developmental dynamics'.

Sincerely

Justin M. O'Sullivan

Reviewer #1:

- The authors should comment on their rationale to include only common SNP variants but not rare variants and CNV findings.

We have modified the manuscript's limitations section (page 5, line 176) to include the following statement to address this:

"Only common SNPs were included in this analysis as we focused on variants identified as associated with ADHD through GWAS. It would be interesting to expand this work to include CNVs, however this would require different methodology than using spatial eQTLs within the GRNs to elucidate putatively causal mechanisms. Rare SNPs were also ignored as a larger dataset would be required to identify putatively causal genes through 2SMR. It remains possible that rare variants contribute to the regulation of the identified genes, however, further work would need to be done on this."

- Three of the proposed genes (ST3GAL3, PTPRF, TIE1) cluster in a single region on chromosome 1 and are surprising close to each other (within 650 kb). This should be explicitly mentioned and the potential implications should be discussed.

We've modified the manuscript's result section (page 3, line 92) to include the following statement to address this:

"Notably, three of these genes (*ST3GAL3*, *PTPRF*, and *TIE1*) are located on chromosome 1p31, within 650 kb of each other. All three genes are regulated by cis eQTLs which were not in LD with each other, a requirement for inclusion in 2SMR. The 1p31 locus has previously been shown to be associated with psychiatric disorders (Monistrol-Mula et al. 2025) and ADHD specifically (Hall et al. 2021; Deng et al. 2024; Demontis et al. 2023)."

- The link of the candidate genes to immunological mechanisms should be discussed in more detail with respect to the co-morbid involvement of the immune system in ADHD, such as allergies, atopic dermatitis, asthma etc. Likewise the link to metabolic and cardiovascular disease should be given more attention.

We did have a paragraph on both immunological mechanisms and metabolic conditions in the discussion (page 7, line 279). However, in response to the reviewers

suggestion, we have expanded the relevant paragraphs as illustrated below (additions in red):

Önder et al. (Önder, Gizli Çoban, and Sürer Adanır 2021) suggested that inflammatory responses within ADHD were related to the close relationship between ADHD and immune traits, something that we have seen in our analysis, particularly with the link to the autoimmune condition rheumatoid arthritis. However, it is unknown which inflammatory-related genes are responsible for this relationship between inflammation and ADHD (Önder, Gizli Çoban, and Sürer Adanır 2021; Leffa, Torres, and Rohde 2019). The causal gene *PTPRF* has been linked to inflammation through increasing cytokine levels (Huang et al. 2022). *PTPRF* protein interacts with *PPFIA1* (Liprin-alpha-1) which is regulated by SNPs associated with lymphocyte count in both fetal cortical tissue and whole blood. Neutrophil-lymphocyte ratio is significantly higher in individuals with ADHD and has been identified as a potential biomarker for inflammatory mechanisms related to ADHD (Önder, Gizli Çoban, and Sürer Adanır 2021).

Many protein tyrosine phosphatases (PTPs), including *PTPRF*, are differentially expressed in immune cells compared to non-immune cells (Arimura and Yagi 2010). Loss of *PTPRF* in mice has been shown to cause hyperactivation of plasmacytoid dendritic immune cells and mild colitis (Bunin et al. 2015). ADHD has been linked to the immune system previously, both through immune-related genes and the co-occurrence of immune conditions such as allergies and autoimmune diseases (Cortese et al. 2018; Hoekstra 2019). Therefore, we contend that ADHD causal gene *PTPRF* and the interactions between proteins *PTPRF* and *PPFIA1* may be important factors in the interplay between inflammation, the immune system, and ADHD.

Metabolic paragraph (new bits in red):

Metabolic traits were identified as co-occurring with ADHD in both our network analyses and in previous studies (Chen et al. 2018; Huber et al. 2023). However, we did not identify any causal ADHD genes in the liver GRN. This finding supports the hypothesis that ADHD is a neurodevelopmental condition with metabolic co-occurring traits, not the opposite. Despite this, the causal genes are associated with many cholesterol-related traits through pathways in the liver. This is consistent with epidemiological results which have identified a strong link between cholesterol levels and ADHD (Ugur et al. 2018; Xu, Bao, and Liu 2021). Similarly, individuals with ADHD have a heightened risk (OR = 1.96) of developing cardiovascular diseases (CVDs) (Li et al. 2023). In our analysis we did not identify any specific co-occurring CVDs but instead focussed on cholesterol. Garcia-Marin et al. (García-Marín et al. 2021) identified genetic variants associated with low HDL levels and an increased ADHD risk. However, the biological mechanisms that intertwine cholesterol metabolism and ADHD remain unknown. Here, we have identified a potential mechanism for their association through causal gene *ST3GAL3*. In fetal and adult cortical tissue, we have shown *ST3GAL3* to be linked with educational attainment and cognition in individuals with ADHD; whilst in the liver it is associated with *ST3GAL4* which is regulated by many genetic variants associated with cholesterol traits. Therefore, it is proposed that within the brain *ST3GAL3* is related to the cognitive traits linked with ADHD, whilst in the liver it is associated with the cholesterol co-occurring traits.

- Minor points:

page 6, line 231: The reference should be replaced or expanded with Demontis et al. 2019.

Done

page 6, line 239: promoter

Done
page 9, line 383: bootstrapped

Done
Reviewer #2:
No comments to address.

February 11, 2025

RE: Life Science Alliance Manuscript #LSA-2024-03029R

Dr. Justin M. O'Sullivan
The University of Auckland
The Liggins Institute
Private Bag 92019
Auckland 1142
New Zealand

Dear Dr. O'Sullivan,

Thank you for submitting your revised manuscript entitled "Unravelling ADHD: Genes, Co-occurring Traits and Developmental Dynamics". We would be happy to publish your paper in Life Science Alliance pending final revisions necessary to meet our formatting guidelines.

- please be sure that the authorship listing and order is correct
- please upload your main manuscript text as an editable doc file
- please add the Twitter/X and Bluesky handles of your host institute/organization as well as your own or/and one of the authors in our system
- the contributions selected for Evgeniia Golovina; Joerg S Wicker and Jessie C Jacobsen do not qualify them for authorship. Please either update the contributions in our system and the Author Contributions section of the manuscript or let us know if the authors need to be removed (and added eventually to the acknowledgment section)

A. FINAL FILES:

B. MANUSCRIPT ORGANIZATION AND FORMATTING:

Sincerely,

February 14, 2025

RE: Life Science Alliance Manuscript #LSA-2024-03029RR

Dr. Justin M. O'Sullivan
The University of Auckland
The Liggins Institute
University of Auckland
Private Bag 92019
Auckland 1142
New Zealand

Dear Dr. O'Sullivan,

Thank you for submitting your Research Article entitled "Unravelling ADHD: Genes, Co-occurring Traits and Developmental Dynamics". It is a pleasure to let you know that your manuscript is now accepted for publication in Life Science Alliance. Congratulations on this interesting work.

DISTRIBUTION OF MATERIALS:

Again, congratulations on a very nice paper. I hope you found the review process to be constructive and are pleased with how the manuscript was handled editorially. We look forward to future exciting submissions from your lab.

Sincerely,
